# New Urea Controlled-Release Fertilizers Based on Bentonite and Carnauba Wax

João Fernandes Duarte Neto [1], Jucielle Veras Fernandes [2], Alisson Mendes Rodrigues [1,2,*], Romualdo Rodrigues Menezes [1,2] and Gelmires de Araújo Neves [1,2]

[1] Laboratory of Materials Technology (LTM), Department of Materials Engineering, Federal University of Campina Grande, Av. Aprígio Veloso—882, Bodocongó, Campina Grande 58429-900, Brazil
[2] Graduate Program in Materials Science and Engineering (PPG-CEMat), Federal University of Campina Grande, Av. Aprígio Veloso—882, Bodocongó, Campina Grande 58429-900, Brazil
* Correspondence: alisson_mendes@ymail.com

**Abstract:** Controlled-release fertilizers are interesting alternatives to current commercial chemical fertilizers, which present a higher nutrient release rate, and can negatively impact the ecosystem. In this work, two urea controlled-release fertilizer types were manufactured from carnauba wax (CW), commercial granulated urea (U), and natural and sodium bentonite (Bent-R and Bent-Na, respectively). In the first type, the mechanochemical method produced fertilizers in bars, from a mixture containing different proportions of U, Bent-R, and Bent-Na. In the second type, the dip-coating method was used to coat urea bars with coatings containing different proportions of the Bent-R, Bent-Na, and CW. The cumulative urea release was evaluated over the 30-day incubation period, through soil columns tests and UV/visible spectroscopy. Overall, both fertilizers developed in this work presented lower cumulative urea release than standard fertilizers. On the other hand, the new fertilizers produced from the dip-coating method, provided cumulative urea release lower than that obtained by the mechanochemical method. In summary, carnauba wax and bentonite (raw and sodium modified) are promising materials for developing new urea controlled-release fertilizers. Furthermore, both carnauba wax and bentonite are non-toxic, biodegradable, relatively inexpensive, and created from materials that are easily purchased in Brazil, indicating that the new fertilizers developed in this work have the potential to be produced on a large scale.

**Keywords:** bentonite; sodium modified bentonite; controlled-release fertilizers; carnauba wax; mechanochemical method; dip-coating method; urea accumulated release





## 1. Introduction

Currently, the global human population is estimated at 7.8 billion, and it is growing exponentially, with the prospect of reaching 8.5 billion in 2030, and 9.7 billion in 2050 [1]. Because of this, there is a pressing need for agricultural production to increase, to ensure food security. However, such growth must be aligned with other demands of contemporary society, such as sustainability and preserving natural resources [2]. Chemical fertilizers are widely used to increase agricultural productivity, but they present a high leaching rate, leading to several reapplications in short intervals, causing excess nutrients that can cause significant damage to soil and ecosystems, including water pollution and degraded air quality [3]. As such, the controlled leaching rate provided by some fertilizers is beneficial, because it can sustainably increase crop yield in agriculture. Furthermore, controlled-release fertilizers manufactured from natural and sustainable materials, help meet the growing food demand, reduce the number of fertilization applications, contribute to soil preservation, and reduce negative environmental impacts [4,5].

Recently, the scientific community has been working on several methods to develop controlled-release fertilizers, with two methods standing out among them: application of coatings, and adding components to chemical fertilizers. Of the materials used as

coatings to develop controlled-release fertilizers, synthetic and natural polymers stand out. However, the fragility, toxicity, and non-biodegradability of synthetic polymers, and the hydrophilicity of natural polymers, are intrinsic disadvantages of these types of materials [6]. Given this, several studies have been published seeking to develop biodegradable, stable, and economically viable coatings.

Despite this, few works have been directed toward developing such coatings from clays. Among these few works, the publication by Xiaoyu et al. [7] is noteworthy, for developing lower controlled-release fertilizers from bentonite (B) and organic polymers (OP). In this work, bentonite, organic polymer, and a mixture of bentonite and organic polymer, were separately added to molten urea (MU). The leaching solution method was used to evaluate urea release, and the results were compared with commercial urea. All fertilizers developed showed lower urea release than that observed with commercial urea. Assimi et al. [8] studied the leaching behavior of water-soluble diammonium phosphate (DAP) fertilizer, doubly coated, with chitosan-montmorillonite composites as the inner coating and paraffin wax as the outer coating. These authors also reported that the applied coatings were more efficient in reducing nutrient leaching than were uncoated fertilizers. Hermida and Agustian [9] investigated the leaching of urea from fertilizers developed from commercial urea, natural bentonite, and two binders (corn starch and hydroxypropyl methylcellulose). The experimental procedure consisted of adding bentonite and a gel formed from the binders, to the urea, melted at 130 °C. After homogenization via agitation, the substance was poured into a mold and extruded, to produce fertilizer in the form of pellets. Static release experiments demonstrated that the synthesized fertilizers released urea into the water more slowly than did conventional urea fertilizers.

While bentonite already has a history of being used to develop controlled-release fertilizers, the use of carnauba wax to develop such fertilizers has yet to be reported. Carnauba wax has a vegetable origin, extracted from the leaves of a palm tree (Copernicia cerifera) native to northeastern Brazil. The hydrophobic characteristics of this wax are mainly due to the presence of long-chain fatty acids and esters [10]. Due to these properties, carnauba wax is widely used as an additive in various industrial sectors, including cosmetics, pharmaceuticals, and agrochemicals. Indeed, the natural origin of carnauba wax, its easy acquisition in Brazil, low chemical degradation [11], and the fact that it is not toxic to plants or animals make this wax a potential candidate for use as a coating for controlled-release fertilizers. In the case of clays, their intermolecular forces lead to the slow release of some agrochemicals, which results in controlled release periods, thus allowing for the metabolic needs of plants [12,13].

Brazil is responsible for around 8% of global fertilizer consumption, occupying the 4th position, behind only China, India, and the United States. However, more than 80% of the fertilizers used in Brazil are imported [14]. This dependence leaves the Brazilian economy vulnerable to fluctuations in the international fertilizer market. Therefore, developing new controlled-release and sustainable fertilizers from cheap raw materials easily acquired in Brazil [15,16], emerges as an option for reducing Brazil's dependence on international trade. On the other hand, developing controlled-release fertilizers based on urea is justified because nitrogen fertilizers represent 29% of all fertilizers used in Brazilian territory [14]. Therefore, given the above discussion, new urea controlled-release fertilizers have been developed from carnauba wax, commercial urea, and natural and sodium bentonite. Mechanochemical and dip-coating encapsulation methods were used to develop these fertilizers, and the release test in soil columns was used to evaluate the released amount of urea.

## 2. Materials and Methods

### 2.1. Materials

Granulated commercial urea (95%; Heringer, Paulínia, Brazil), raw bentonite (Dolomil Industrial Ltda., Campina Grande, Brazil), carnauba wax (CW)—type 3 (Altos Ceras, Teresina, Brazil), sodium carbonate (Vetec, Duque de Caxias, Brazil), 4-dimethylamino

benzaldehyde (Dinâmica Química Contemporânea Ltda., Indaiatuba, Brazil), hydrochloric acid (Vetec Química, Duque de Caxias, Brazil), and trichloroacetic acid (Dinâmica Química Contemporânea Ltd., Indaiatuba, Brazil).

## 2.2. Preparation of the Sodium-Modified Bentonite

The crude bentonite was disintegrated in a disc mill (Marconi model MA-700, Piracicaba, Brazil), and sieved (0.074 mm) with a mechanical sieve (Vibrotec CT-025, Tubarão, Brazil). The sodium modification was carried out using sodium carbonate ($Na_2CO_3$) to change the "exchangeable cations" of clay for sodium ($Na^+$). This modification followed the methodology used for bentonite in drilling fluids [17], intending to acquire a cation exchange of 100 meq/100 g of clay. Therefore, 6.44 mL of 1.0 mol/L $Na_2CO_3$ solution (100 meq of $Na_2CO_3$) was added to 24.3 g of clay. To ensure a homogeneous mixture, distilled water was added. After this preparation, the samples were placed in plastic containers for five days. The samples were kept at rest for five days. The sample was identified as Bent-Na, and the raw bentonite was called Bent-R throughout the work.

## 2.3. Materials Characterization

The Bent-Na and Bent-R samples sieved (75 μm) were characterized by particle distribution techniques via low-angle laser light scattering (CILAS-1064, Orléans, France). The identification of mineralogical phases was accomplished by X-ray Diffraction (XRD 6000-Shimadzu, Kyoto, Japan) with Kα radiation from copper (Cu), voltage/current of 40 KV/30 mA, the step of 0.02°, and 0.6 s per step. The thermal behavior was evaluated by thermogravimetric analysis (TG), and differential thermal analysis (DTA, Shimadzu-TA60, Kyoto, Japan), performed under a heating rate of 5.0 °C/min, up to a maximum temperature of 1000 °C.

## 2.4. Synthesis of the Urea Controlled-Release Fertilizers

### 2.4.1. Mechanochemical Method

Figure 1 shows the experimental scheme for producing urea-controlled-release fertilizers using the mechanochemical method. Two mixtures were prepared, the first containing granulated urea + Bent-R, and the second containing granulated urea + Bent-Na, in proportions 1:1, 2:1 (66.6% urea), and 4:1 (80% urea). The mixtures were homogenized in a parakeet mill (Servitech model CT-12242, Tubarão, Brazil) at 374 rpm for 15 min. Then, these compositions were dry pressed (2.0 ton/20 s) (Ribeiro-RP0002, Bom Jesus dos Perdões, Brazil) to produce the fertilizers in rectangular bars (30 mm × 5 mm × 5 mm). The terminology used for all fertilizers synthesized in this work can be seen in Table 1.

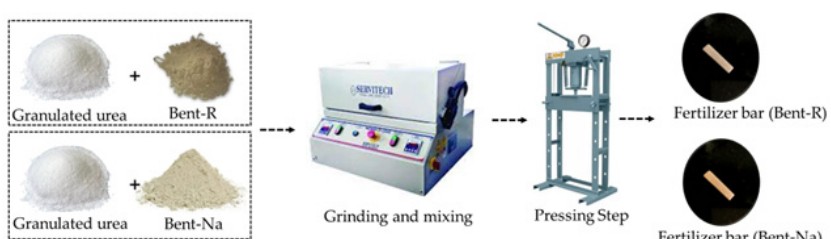

**Figure 1.** Experimental scheme used to develop urea controlled-release fertilizers by the mechanochemical method.

**Table 1.** Nomenclature of the urea controlled-release fertilizers synthesized using the mechanochemical method.

| Nomenclature | Compositions (wt%) | | |
| --- | --- | --- | --- |
| | Urea | Bent-R | Bent-Na |
| U | 100% | - | - |
| M1 | 50% | 50% | - |

**Table 1.** *Cont.*

| Nomenclature | Compositions (wt%) | | |
| --- | --- | --- | --- |
| | Urea | Bent-R | Bent-Na |
| M2 | 67% | 33% | - |
| M3 | 80% | 20% | - |
| MNa1 | 50% | - | 50% |
| MNa2 | 67% | - | 33% |
| MNa3 | 80% | - | 20% |

2.4.2. Dip-Coating Method

Figure 2 summarizes the experimental scheme to develop coated urea fertilizers. Step 1—Preparation of the urea bars: the granulated urea was ground for 10 min (Servitech model CT-12242, Brazil). Then, 2 g of the material was pressed (2.0 ton/20 s, Ribeiro-RP0002) to produce the fertilizers in a rectangular format (30 mm × 5 mm × 5 mm). Step 2—Synthesis of coating solutions: the carnauba wax was melted at 83 °C; after the melting time, the Bent-R and the Bent-Na contents were added in separate CW solutions with pre-established concentrations, (Table 2) and remained under mechanical agitation for 1 h. Step 3—Fertilizer coating: the urea bars were immersed twice in the solutions prepared in step 2 (~10 s for each immersion), to produce the coating, and then dried at room temperature. Table 2 shows the terminology used for coated fertilizers and the ratios of CW, Bent-R, and Bent-Na, used to obtain their respective coatings.

**Step 1: urea bars preparation**

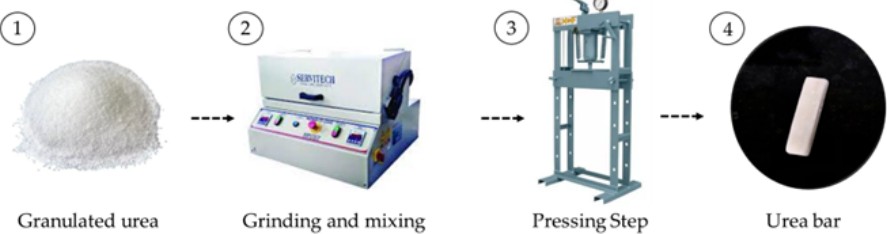

**Step 2: synthesis of coating solutions**

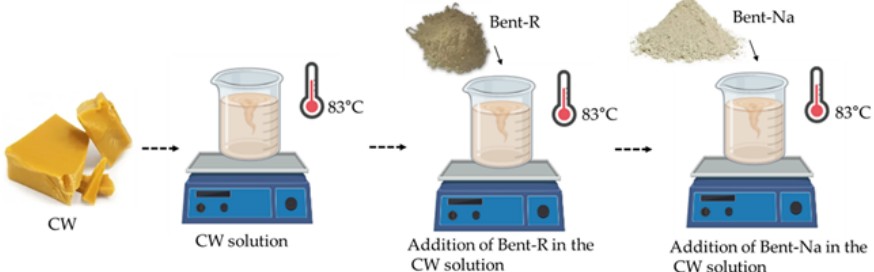

**Step 3: coated urea fertilizers**

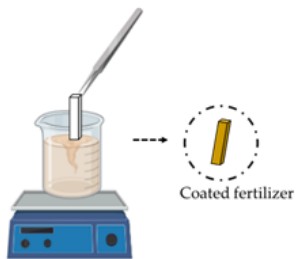

**Figure 2.** The experimental scheme used to produce urea controlled-release fertilizers using the dip-coating method: (1) preparation of urea bars by pressing; (2) synthesis of the coating solution; and (3) immersion of urea bars into the coating solution.

**Table 2.** Nomenclature of urea controlled-release fertilizers produced by the dip-coating method, and the proportions of CW, Bent-R, and Bent-Na, used to obtain the coatings.

| Nomenclature | Compositions (wt%) | | | |
|---|---|---|---|---|
| | **Urea** | **Bent-R** | **Bent-Na** | **CW** |
| UC | | | | 100% |
| UCB10 | | 10% | | 90% |
| UCB40 | | 40% | | 60% |
| UCB50 | Urea bars | 50% | | 50% |
| UCB60 | | 60% | | 40% |
| UCBNa10 | | | 10% | 90% |
| UCBNa40 | | | 40% | 60% |
| UCBNa50 | | | 50% | 50% |
| UCBNa60 | | | 60% | 40% |

### 2.5. Soil Column Tests

The soil column tests assessed urea release from all urea controlled-release fertilizers synthesized in this work. To accomplish this, PVC tubes (7.5 cm in diameter and 30 cm high) were closed, allowing a 0.6 cm diameter hole to collect the percolated solution. A sterile gauze pad and filter paper were placed on the bottom, to prevent soil loss and to filter the percolated solution. Each tube was filled with 1.3 kg of soil, up to a height of 25 cm. The soil sample was sieved (2 mm) and dried for 48 h at 110 °C, for the urease enzyme denaturation. In each PVC column, soil samples presented similar densities to those in the environment (~1.18 g·cm$^{-3}$). The urea controlled-release fertilizers were inserted every five centimeters in column depth. Figure 3 outlines how the ground column test was carried out. The soil columns were irrigated 10 times over 30 incubation days. An amount of 400 mL of the water was used in the first irrigation, and 75% was retained in the soil column. For the second irrigation, 100 mL of water was used. The irrigations were carried out every 72 h, with aliquots of the percolated solution removed every 48 h after the irrigations.

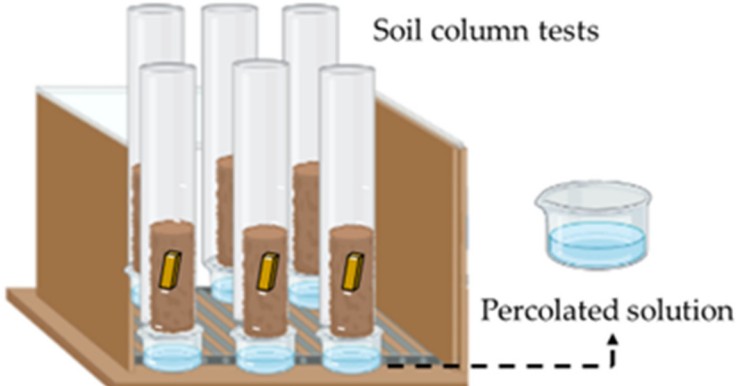

**Figure 3.** Scheme of the soil column test.

The urea concentration was determined using the UV-visible spectrophotometer (UV-1800 Shimadzu Spectrophotometer, Kyoto, Japan), in the wavelength range of 420 nm. The methodology for detecting urea in the UV-visible spectrophotometer, consisted of preparing the Ehrlich reagent (0.36 mol·L$^{-1}$ solution of 4-dimethylaminobenzaldehyde in 2.4 mol·L$^{-1}$ HCl) + a solution of 10% trichloroacetic acid. The dissolution of urea was evaluated by analyzing the concentrations of the product of the reaction of urea with Ehrlich's reagent. The total amount of urea leached during the month, was obtained by adding the leached quantities in each irrigation.

### 3. Results

#### 3.1. Characterization of Clays

The particle size distribution analysis of Bent-R and Bent-Na is summarized in Figure 4a,b. The Bent-Na and Bent-R samples presented bimodal and monomodal behaviors, respectively. The bimodal behavior of the Bent-Na sample is due to the presence of sodium ions ($Na^+$) as an exchangeable cation in the clay mineral interlayer, which causes greater deagglomeration between the particles of the clay fraction [18]. The particle size distributions were 0.1–50 μm and 1–50 μm for Bent-Na and Bent-R, respectively.

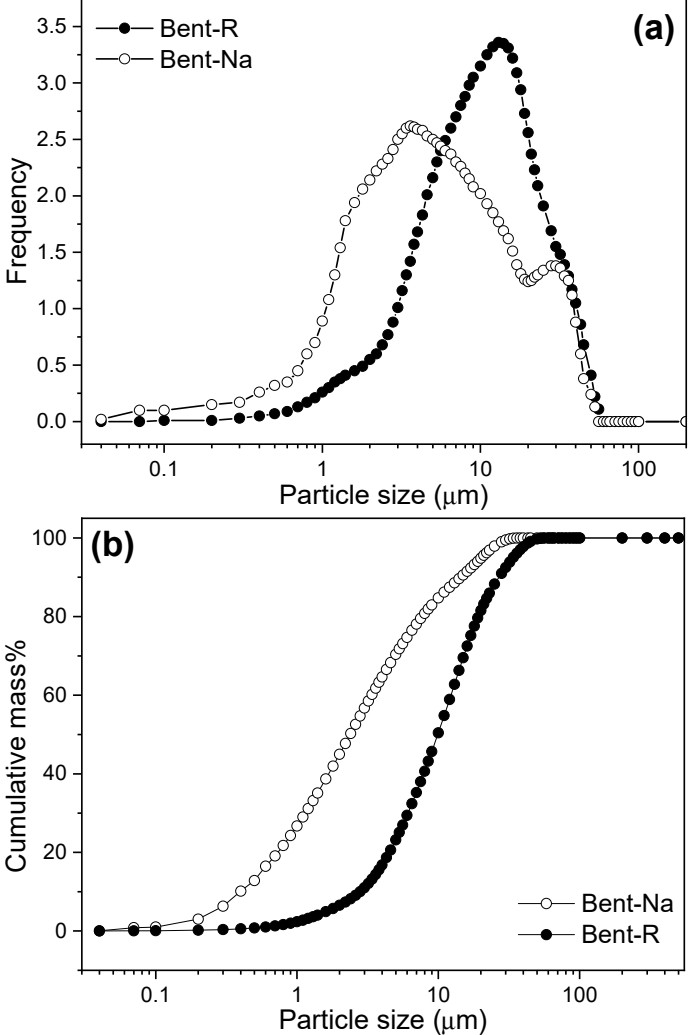

**Figure 4.** (**a**,**b**) Particle size distribution curves of the Bent-R and Bent-R clays.

The chemical analysis measured from the Bent-Na and Bent-R samples is summarized in Table 3. As expected, both clays presented $SiO_2$ and $Al_2O_3$ as major constituents, corresponding to more than 60% of their chemical composition [19–22]. The detected $SiO_2$ comes from the tetrahedral layer of the smectite clay mineral and quartz, while the $Al_2O_3$ is present in the octahedral layer of the clay mineral. The presence of $Fe_2O_3$ is associated with the isomorphic substitution of $Al^{3+}$ ions by $Fe^{3+}$ in octahedral sites, as well as in the form of hydroxides. Bent-R is polycationic with the presence of $Ca^{2+}$ and $Mg^{2+}$ ions. The 20.6% fire loss for Bent-R is probably related to moisture, coordinated and adsorbed water losses, organic matter burning, carbonate decomposition, and structural destruction of clay minerals. In general, it was observed that Bent-R has a typical chemical composition of bentonite clays [23,24]. Bent-Na presented all constituents detected in natural clay; the sodium present resulted from sodium functionalization.

**Table 3.** Chemical composition of Bent-R and Bent-R clays.

| Samples | SiO$_2$ | Al$_2$O$_3$ | F$_2$O$_3$ | CaO | Na$_2$O | MgO | K$_2$O | Other | FL [1] |
|---------|---------|-------------|------------|------|---------|------|--------|-------|--------|
| Bent-R  | 47.4%   | 15.3%       | 9.2%       | 3.2% | ND      | 2.3% | 0.5%   | 1.4%  | 20.6%  |
| Bent-Na | 48.6%   | 15.5%       | 10.1%      | 3.3% | 1.5%    | 2.3% | 0.5%   | 1.5%  | 16.7%  |

[1.] Fire loss.

The mineralogical phases in Bent-R and Bent-Na were identified from the diffractograms shown in Figure 5. In both, the clay minerals smectite (JCPDS 10-0357), quartz (JCPDS 46-1045), and calcite (JCPDS 47-1743), were identified. In Bent-R, the interplanar distances were identified: for the clay mineral smectite, at 15.51 Å, 4.50 Å, and 2.55 Å; for the quartz, at 4.25 Å, 3.34 Å, 2.45 Å, and 1.82 Å; and for the calcite, at 3.02 Å and 2.28 Å. The same distances were identified in Bent-Na, except for the main peak of smectite d$_{001}$, which presented an interplanar distance of 12.50 Å. These results agree with the chemical composition shown in Table 3, and also with previously published studies [25–27]. In addition to the difference in interlamellar spacing, the intensity of the main peak d$_{001}$ of smectite in Bent-Na was also lower [28,29]. This effect is probably related to the water amount in the clay mineral interlayers. Bent-Na showed smaller amounts of water, as seen in the lower fire loss (Table 3).

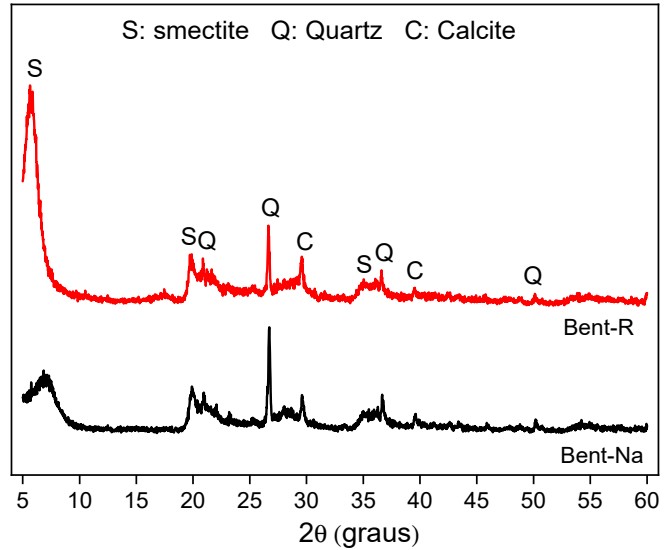

**Figure 5.** Diffractograms obtained from Bent-R and Bent-Na samples.

The thermal behaviors of Bent-R and Bent-Na clays were investigated via TG experiments and your respective derivative (Figure 6a,b). From the TG curves, for both clays, 4 thermal events were identified: 70–195 °C, 196–312 °C, 313–570 °C, and 620–700 °C. The total mass loss was 22.02% and 17.90%, for the Bent-R and Bent-Na samples, respectively. The first thermal event detected the most significant mass loss (15.32% and 9.7% for Bent-R and Bent-Na, respectively). This sharp mass loss is related to the loss of hydration, adsorbed, and coordinated waters [30]. The mass loss between 196–312 °C is related to iron hydroxide dihydroxylation. Another significant mass loss occurred between 313–570 °C, and is related to the dihydroxylation of clay minerals. This event occurs at temperatures lower than those indicated for the montmorillonite clay mineral, due to iron in the tetrahedral and octahedral sheets of the clay minerals [31]. The final mass losses were detected between 620–700 °C in both samples, with their origins related to the decomposition of carbonates. In this temperature range, the mass loss of the Bent-Na sample (2.4%, see Figure 6b) was more significant than the mass loss of the Bent-R sample (2.05%, see Figure 6a), due to the presence of Na$_2$CO$_3$ in its composition [28]. These data corroborate with the results

obtained in the chemical composition and the XRD analyses, which indicate the presence of $CaCO_3$.

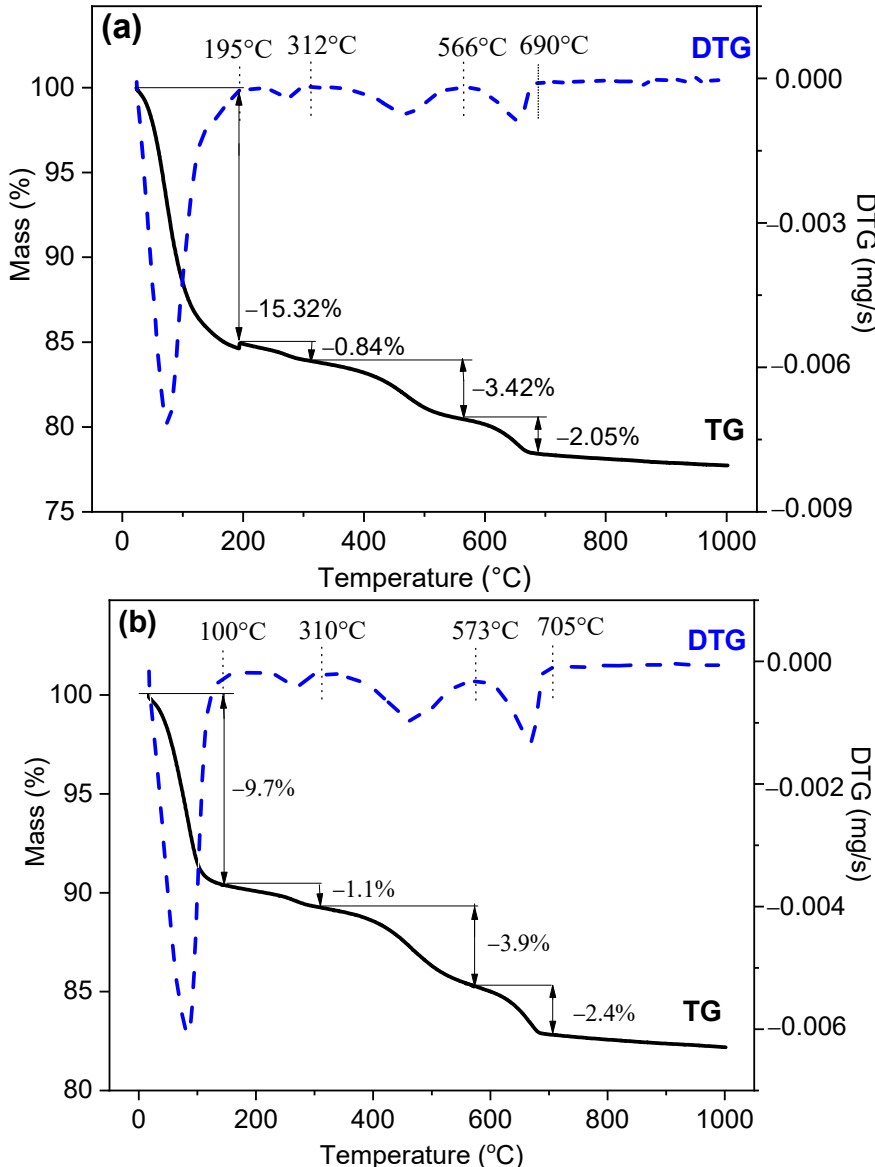

**Figure 6.** TG/DTG curves of (**a**) Bent-R (**b**) Bent-Na clays.

### 3.2. Cumulative Urea Release Analysis

The urea-leaching from the new fertilizers produced in this work was evaluated over 30 days, using the soil columns tests. Figure 7a compares the cumulative urea release as a function of the incubation days of a standard fertilizer (commercial urea) and the new urea controlled-release fertilizers produced using the mechanochemical method. In this work, $t_{UCR}^{100\%}$ represented when the cumulative urea release time reached 100%. For the standard fertilizer (U), $t_{UCR}^{100\%}$ was equal to 12 days. On the other hand, the M1, M2, and M3 fertilizers reached $t_{UCR}^{100\%}$ at 18 days. For the MNa1, MNa2, and MNa3 fertilizers, $t_{UCR}^{100\%}$ were equal to 15 days. These results indicate that the strategy of adding both raw and sodium bentonite to granulated urea, was effective in increasing $t_{UCR}^{100\%}$ values. On average, the fertilizers produced by adding sodium bentonite showed higher $t_{UCR}^{100\%}$ values than those obtained by adding raw bentonite. This behavior can be explained because sodium bentonite enables excellent water retention in the granule. The $Na^+$ ions allow for greater water interleaving in the interlayer spaces, resulting in increased hydration and, therefore,

increased spacing between the layers of the clay mineral. This sodium bentonite hydration prevents immediate urea solubilization, even if only for a short period [32].

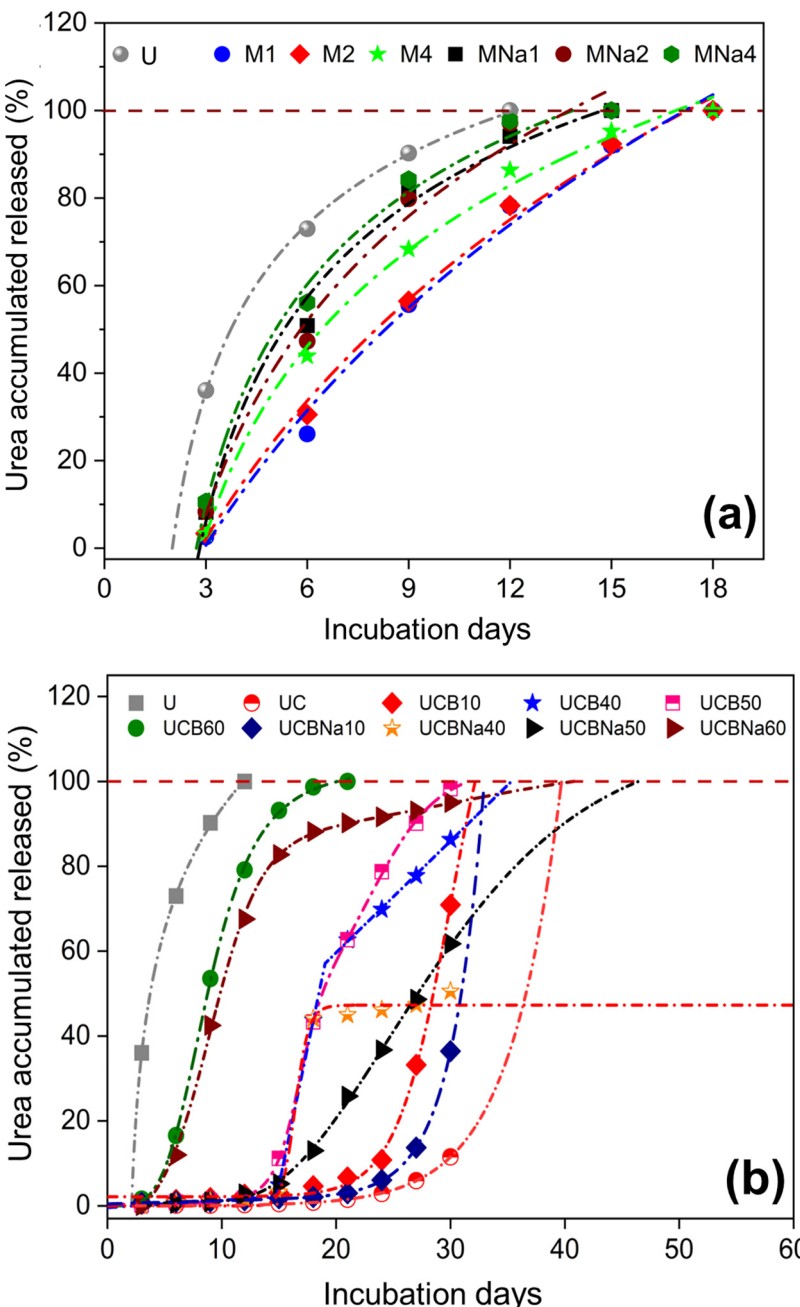

**Figure 7.** Cumulative release of urea as a function of incubation days for (**a**) urea controlled-release fertilizers produced by the mechanochemical method and (**b**) urea controlled-release fertilizers produced by the dip coating method.

Figure 7b shows the cumulative urea release as a function of incubation days, for the U fertilizer coated only with CW (UC), U coated with CW + Bent-R (UCB10, UC40, UCB50, and UCB60), and U coated with CW + Bent-Na (UCBNa10, UCBNa40, UCBNa50, and UCBNa60). The UCB50 and UCB60 fertilizers presented $t_{UCR}^{100\%}$ values equal to 31 and 20 days, respectively. At 30 days, the cumulative urea release values were equal to 11.5%, 71%, 86%, 36.45%, 47.3%, 61.4%, and 95% for UC, UCB10, UCB40, UCBNa10, UCBNa40, UCBNa50, and UCBNa60, respectively. In this case, mathematical adjustments were performed to determine $t_{UCR}^{100\%}$ calculated values for these fertilizers. After this analysis, it was

possible to conclude that those fertilizers coated with carnauba wax + Bent-Na, presented lower cumulative urea released urea than fertilizers coated with carnauba wax + Bent-R. Therefore, with the coated fertilizers, Bent-Na clays were more effective at increasing $t_{UCR}^{100\%}$ than were Bent-R clays. CW-encapsulated fertilizers were more efficient than uncoated fertilizers. However, UCBNa60 and UCB60 experienced the undesirable "explosion effect" (coating rupture), as they showed trends of rapid urea release in the initial phase of the experiment. The coated samples inhibit the release of urea for a long time due to the hydrophobic character of carnauba wax, reducing the diffusivity of water molecules and preventing the solubilization of urea and subsequent leaching. In this sense, the surface coated with CW exerts a mechanism by repelling the water that tries to penetrate inside, where urea is found, thus delaying the solubilization of the nutrient. On the other hand, Bentonite can act by competing for the water that manages to penetrate the wax, preventing the urea from solubilizing quickly in a short period. Table 4 compares experimental and calculated $t_{UCR}^{100\%}$ values for all urea controlled-release fertilizers developed in this work.

**Table 4.** Experimental and calculated $t_{CRU}^{100\%}$ values for all urea controlled-release fertilizers developed in this work.

| Dip-Coating | $t_{UCR}^{100\%}$ (Days) | Mechanochemical | $t_{UCR}^{100\%}$ (Days) |
|---|---|---|---|
| UC | indeterminate | U | 12 |
| UCB10 | 39.5 | M1 | 18 |
| UCB40 | 35 * | M2 | 18 |
| UCB50 | 30 | M3 | 18 |
| UCB60 | 20 | MNa1 | 15 |
| UCBNa10 | 33 * | MNa2 | 15 |
| UCBNa40 | 51 * | MNa3 | 15 |
| UCBNa50 | 46 * | - | - |
| UCBNa60 | 41 * | - | - |

\* $t_{UCR}^{100\%}$ values calculated.

The leaching of urea in sandy soil type (used in this research) tends to be significantly faster, indicating no urea retention or by-products. This hypothesis is reasonable, considering that this process is already known in the literature on soils. It is worth noting that, depending on the soil type and the fertilizer application model, these can decisively influence the release behavior, limiting or suppressing the effects of controlled release systems.

## 4. Discussion

The cumulative urea released from the new fertilizers developed in this work was discussed for 1 and 30 incubation days, and compared with literature data. For the 30 day incubation period, the discussion was carried out regarding experimental and calculated cumulative urea release values (see Figure 7a,b). As for the 1-day incubation period, only calculated values were taken into account. Figure 8 compares the cumulative urea releases over 30 incubation days of the new urea controlled-release fertilizers developed in this study, with others reported in the literature (hatched columns). All urea controlled-release fertilizers produced by the mechanochemical method (M1, M2, M3, MNa1, MNa2 and MNa3), reached the 30th day of the experiment with 100% of the urea present leached. Indeed, as shown in Table 4, the $t_{UCR}^{100\%}$ values for these fertilizers were reached well before the 30th day of the experiment.

As for the urea controlled-release fertilizers produced by the dip-coating method, only the UCB60 fertilizer showed an accumulated release of urea equal to 100% on the 30th day. In the literature, it is possible to find fertilizers that presented similar behavior, for example, BCU3% [33], SBCU3% [33], EC/SPC [34], SRF/HNTS [35], and e SRF [35]. The UCB50 and UCBNa60 fertilizers showed accumulated urea release values above 90% on the 30th day (97.95% and 93.83%, respectively). Such behavior is similar to CUF [13], NIUKF [13], and e SSBCU3% [33] (99%, 98.15%, and 90%, respectively). The UCB40, UCB10, UCBNa50,

UCBNa40, UCBNa10, and UC fertilizers, presented accumulated urea release values that were lower than 90% in 30 days of the experiment (86.52%, 71.0%, 61.4%, 47.3%, 36.2%, and 11.5%, respectively). From Figure 6, it is possible to verify that the following fertilizers reported in the literature also presented similar values of the accumulated release of urea: EC/NS, UIKF [13], PUC1-1 [36], PUC1-2 [36], e CMCK-g-P [37] (81%, 70%, 70%, 54%, and 2.5%, respectively). It is worth mentioning that the UCBNa40 fertilizer showed the lowest accumulated urea release value on the 30th incubation day. Furthermore, the mathematical adjustment indicated that the urea released after the 30th incubation day is insignificant, indicating that such fertilizer has no effective action after this.

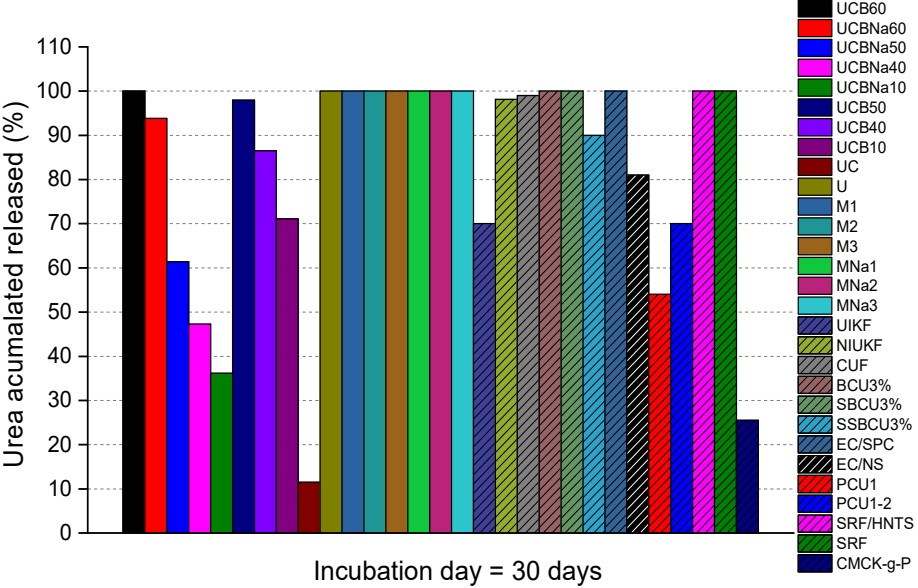

**Figure 8.** Compares the measured and calculated cumulative urea release time at 30 days of the urea controlled-release fertilizers developed in this work.

The urea cumulative release for 1 incubation day was below <1% for all urea controlled-release fertilizers developed in this work (see Figure 7). This is significantly lower than some fertilizers reported in the literature, such as the works by Mahdavi et al. [13] (UIKF, NIUKF, and CUF fertilizers, that release 13.7%, 25.6%, and 80.9% of urea, respectively); Zhang et al. [33] (BCU3%, SBCU3%, SSBU3%, EC/SPC, EC/NS fertilizers that release 79.9%, 15.9%, 5.9%, 26.9%, and 8.4% of urea, respectively); and Shen et al. [35] (SRF/HNTS and SRF fertilizers that release 61.2% and 52.9% of urea, respectively). On the other hand, the works published by Liu et al. [36] (PCU1-1 and PC1-2), and Wang et al. [37] (CMCK-g-P), also did not show a significant release of urea for 1 incubation day. However, they are fertilizers that require greater complexity in their synthesis.

The fertilizers formulated in this work present values similar to those found in the literature, with the UC sample from our work being the one that released the lowest percentage of urea (less than 15% in 30 days). This shows that the fertilizers formulated in this research have the potential to be used as slow/controlled-release fertilizers, since they have greater longevity for urea release, when compared to commercial urea and even to other controlled-release fertilizers found in the literature, as seen in Figure 8. However, using these samples will depend on the crop type and soil. Precision agriculture aims to ensure that plants have access to the necessary nutrients at all stages of their growth: germination, shoot formation, flowering, grain filling, and fruit formation [38]. Therefore, each crop type has these phases at different times, depending on factors such as sowing time, weather conditions, soil type, and crop rotation. Given these factors, different types of controlled-release fertilizers will suit each crop type.

## 5. Conclusions

New urea controlled-release fertilizers were successfully synthesized using mechanochemical and dip-coating methods, incorporating commercial urea, carnauba wax, and natural and sodium bentonite. From the soil column tests, it was possible to conclude that all the fertilizers produced by the mechanochemical method, exhibited a lower urea release rate than that of standard fertilizer. The fertilizers coated with CW + Bent-R and CW + Bent-Na, retained urea for longer, as evidenced by higher $t_{UCR}^{100\%}$ values than those obtained by using the mechanochemical method. Among the dip-coating fertilizers, only UCB50 and UCB60 presented experimental $t_{UCR}^{100\%}$ values, of 31 and 20 days, respectively. However, experimental $t_{UCR}^{100\%}$ values could not be obtained for UCB10, UCB40, UCBNa10, UCBNa40, UCBNa50, and UCBNa60, and in these cases, mathematical adjustments were made to determine $t_{UCR}^{100\%}$ calculated values.

Overall, this study contributes to developing new, sustainable, cost-effective fertilizers, for improving crop yields and reducing environmental pollution. The easy execution of the methods used to produce these new fertilizers would benefit small, medium, and large sized companies for large-scale production. Furthermore, the fact that they are synthesized from non-toxic and biodegradable materials, adds to their sustainability and potential for use in agriculture.

## 6. Patents

Our research resulted in a deposited patent entitled "Desenvolvimento de Fertilizantes à Base de Bentonita e Cera de Carnaúba para Liberação Controlada de Uréia" BR 102021018233 4 [39].

**Author Contributions:** All authors contributed to achieve this manuscript. J.F.D.N.: sample collection, preparation and lab analysis, data curation, methodology and investigation. J.V.F. and A.M.R.: validation, data curation, editing and article review. R.R.M. and G.d.A.N.: supervision, conceptualization and article review. All authors have read and agreed to the published version of the manuscript.

**Funding:** This research was funded by Coordenação de Aperfeiçoamento de Pessoal de Nível Superior (88887.597478/2021-00) and by Conselho Nacional de Desenvolvimento Científico e Tecnológico (140211/2021-7). RRM is grateful to Conselho Nacional de Desenvolvimento Científico e Tecnológico (Grant number 309771/2021-8). AMR also is grateful to Conselho Nacional de Desenvolvimento Científico e Tecnológico (Grant number 313616/2020-5) and Fundação de Apoio à Pesquisa do Estado da Paraíba (Grant number 48332.712.29500.30082021).

**Institutional Review Board Statement:** Not applicable.

**Informed Consent Statement:** Not applicable.

**Data Availability Statement:** Not applicable.

**Acknowledgments:** The authors would like to thank the Laboratório de Tecnologia dos Materiais, for the support it provided.

**Conflicts of Interest:** The authors declare no conflict of interest.

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
