# Peer review of "New Urea Controlled-Release Fertilizers Based on Bentonite and Carnauba Wax"

_sustainability, doi:10.3390/su15076002_

Round 1

Reviewer 1 Report

1: Abstract need to be revised grammatically. Objective of the research is not mentioned in abstract. The abstract should state briefly the purpose of the research, the principal results and major conclusions. An abstract is often presented separately from the article, so it must be able to stand alone. This section isn't clear. Authors just collecting some ideas. Please, try to improve this section by highlighting the research gap and the novelty of this work. Also, try to lead the reader smoothly to your point.

2: Page 1, Line 36-39: Please revise the statement. Presently it is not clear.

3:Page 2, Line 45-62: Authors have discussed one example for hybrid fertilizer and one for coated fertilizer but to me this much literature is incomplete. Please improve your literature review. Add the below mentioned 05 recent references… (highly recommended)
https://doi.org/10.3390/polym13224040
https://doi.org/10.1016/j.scitotenv.2022.157417%EF%BB%BF

4: Page 8, line 214-221; at authors states "it was impossible to determine ????100% values for the UC, 214 UCB10, UCB40, UCBNa10, UCBNa40, UCBNa50 and UCBNa60 fertilizers since, at the end 215 of the experiment, there was still urea retained." But later they shows the percentage of urea released. I am not getting if they don't know the exact percentage of the urea of their fertilizer how they can decide the percentage released. I suggest authors to use "Kedaljhl method" to find the total available nitrogen content of their fertilizer then they can calculate the release percentage of their fertilizer.

Author Response

Attached answer.

Reviewer 2 Report

Dear Editor

Thank you for inviting me for reviewing a MS entitled New hybrid urea-controlled release fertilizers based on bentonite and carnauba wax. It seems an important contribution on developing synthesized fertilizer. However, I would suggest to consider the following points to be considered before publication.

Abstract

A sentence regarding implication of the study would be essential to write as the final sentence of the manuscript.

Introduction

Line 32 I think the authors mean to write human population.

Line 45-62 I think it is better to highlight the context of the study rather than highlighting the author, …leaching of urea from hybrid fertilizer was analyzed by Hermida and Agustian [4]

Line 64 as far as we know….it’s a very loose sentence you can mention little is known about….

At least a couple of sentences are requires at the end of introduction about the objectives of the study.

Results

Figure captions and legends consist of bigger font size, better to reduce size

Discussion

The discussion section has to be revised, I think the same sub-headings used in results would be ideally fine to write discussion

Characterization of clay

Urea leaching

Some comparisons of present study and related studies are required to be written here. I have seen some literature reviews in the introduction section. So would like to suggest authors to take these points and rewrite the discussion

Is Figure 7 really required in discussion? I think it could be adjusted in result section somewhere, please think of it.

Conclusion

Again, please write the implication of the study here, the work already has patent, why not elaborate on the beneficiary of such studies, implication and further recommendation on synthesis of artificial fertilizers.

Author Response

Attached answers

Reviewer 3 Report

This study tried to develop hybrid fertilizers based on bentonite and carnauba wax and evaluated the release of urea. I recommend this MS to be published after a minor revision.

1.      The abstract should be well improved. Firstly, it may be better to present the significance or contribution or purpose of this study. Additionally, the abstract should focus on critical conclusions or findings of this study instead of methods or simple results.

2.      References were lacking in the introduction, especially the first two paragraphs. The third paragraph only simply presented some papers studying on the development, evaluation, and use of hybrid fertilizers but not concluded their limitation and not further link them to your research to explain the purpose of your research.

3.      The description on results (section 3) is a little wordy, while the in-depth discussion (section 4) on these results and corresponding supporting evidence/references seem to be inadequate.

4.      The conclusions suffered the same problems with the abstract.

Author Response

Attached answers

Reviewer 4 Report

The article looks good, however, it needs some revisions as follows:

1. The authors must describe well about the novelty of the study. There are already many proven options to make slow-release urea. Neem-coated urea is very much popular and most of the urea is now neem-coated. Then what is the need for making new controlled-release urea coating with bentonite and carnauba wax?

2. L 33-34: Use the following reference citing this statement: https://doi.org/10.3390/agronomy11112190 

3. L 75: Above discussion does not signify the development of new hybrid fertilizer, urea. 

4. Clearly mention the novelty statement in the last paragraph of the introduction part. Also, mention the hypothesis followed in this study.

5. There is no mention of the place of this study. The authors must mention it in the materials section.

6. Na-modified bentonite preparation part must be elaborated. If there is a similarity issue, the authors are advised to include this in the supplementary part. 

7. Methodology part is not adequate. How did the author characterize the clays?

8. Discussion part is well-written.

In my view, the article is has the potential to get published. However, the author must state the novelty of the study and a clearcut illustration of the methodology followed.

Author Response

Attached answers

Round 2

Reviewer 4 Report

The authors did a good job to revise the article and in my view, the article may be accepted, now.